# Imaging high-speed friction at the nanometer scale

Per-Anders Thorén[1], Astrid S. de Wijn[2,3], Riccardo Borgani[1], Daniel Forchheimer[1] & David B. Haviland[1]

Friction is a complicated phenomenon involving nonlinear dynamics at different length and time scales. Understanding its microscopic origin requires methods for measuring force on nanometer-scale asperities sliding at velocities reaching centimetres per second. Despite enormous advances in experimental technique, this combination of small length scale and high velocity remain elusive. We present a technique for rapidly measuring the frictional forces on a single asperity over a velocity range from zero to several centimetres per second. At each image pixel we obtain the velocity dependence of both conservative and dissipative forces, revealing the transition from stick-slip to smooth sliding friction. We explain measurements on graphite using a modified Prandtl–Tomlinson model, including the damped elastic deformation of the asperity. With its improved force sensitivity and small sliding amplitude, our method enables rapid and detailed surface mapping of the velocity dependence of frictional forces with less than 10 nm spatial resolution.

[1] Nanostructure Physics, Royal Institute of Technology (KTH), Albanova, SE-10791 Stockholm, Sweden. [2] Department of Physics, Stockholm University, 106 91 Stockholm, Sweden. [3] Department of Engineering Design and Materials, Norwegian University of Science and Technology, 7491 Trondheim, Norway. Correspondence and requests for materials should be addressed to P.-A.T. (email: pathoren@kth.se) or to A.S.d.W. (email: dewijn@fysik.su.se or astrid@dewijn.eu) or to D.B.H. (email: haviland@kth.se).

Many applications in tribology require an understanding of frictional forces on nanometer-scale contacts[1,2] moving with a velocity of at least $1\,cm\,s^{-1}$. Traditional nanoscale friction experiments use an atomic force microscope (AFM), where the frictional force on the tip or colloidal probe is measured while sliding on a surface at constant velocity[3–7]. Friction induces a lateral force on the tip, resulting in a twist $\phi$ around the major axis of the AFM cantilever (see Fig. 1a), which is detected by optical beam deflection. The cantilever's restoring torque is assumed to be in quasi-static equilibrium, in which case the cantilever twist gives the instantaneous lateral force on the tip. When measuring individual stick-slip events with this method[8], one typically neglects cantilever inertia and damping, a valid approach if these events occur at low enough frequency.

The quasi-static method is typically limited by detector noise, where the unity signal-to-noise ratio in a 1 ms measurement time defines a typical minimum detectable force $F_{min} \sim 13\,pN$ (see Methods). With the quasi-static method, individual stick-slip events can be resolved[9] up to velocities $\sim 3\,\mu m\,s^{-1}$, at least four orders of magnitude below the velocity scale relevant to applications. At higher velocity stick-slip events can not be resolved, only the mean force of sliding friction. Scan velocities as high as $580\,\mu m\,s^{-1}$ have been reached[10], but at this velocity a measurement time of 1 ms would limit spatial resolution to 580 nm.

In contrast, dynamic methods sense frictional force as a perturbation to the cantilever's free linear dynamics near a high-quality factor resonance. The high frequency of a stiff torsional resonance $f_0 \sim 2\,MHz$ allows for a maximum tip velocity $v_{max} = 2\pi f_0 A \sim 6\,cm\,s^{-1}$ with very small amplitude of sliding oscillation $A \sim 5\,nm$. Owing to the enhanced force sensitivity of the high $Q$ resonance, a good AFM can see the thermal random torque acting on the cantilever, which is resolved near resonance as twisting Brownian motion noise, above the voltage noise floor of the detector. In this case, force measurement is at the thermal limit of sensitivity, which for the stiff 2 MHz cantilever gives $F_{min} = 0.88\,pN$ in the same 1 ms measurement time (see Methods). While dynamic friction has been previously studied using flexural[11] and torsional[12–14] resonance, thus far dynamic methods have not been used to measure frictional force, only changes of oscillation amplitude and phase when the tip engages a surface.

Here we extend the force measurement methodology of intermodulation AFM[15,16] to lateral forces which are important for understanding friction. We describe a calibrated and quantitative dynamic method of measuring frictional force. At every image pixel, we observe the transition from stick-slip to smooth sliding friction as a characteristic shape in the amplitude dependence of the dynamic force quadratures $F_I(A)$ and $F_Q(A)$. In contrast to the quasi-static method, dynamic force quadratures do not give the instantaneous lateral force on the tip, but rather the conservative force $F_I$ and dissipative force $F_Q$, integrated over one single oscillation cycle of the tip with amplitude $A$ (see Methods).

## Results

**Measurements.** Intermodulation AFM is based on the detection of high-order frequency mixing (intermodulation) near a mechanical resonance. In this work the first torsional eigenmode (a linear oscillator) is driven at two frequencies near resonance. When perturbed by the nonlinear frictional force, the resonator responds with a frequency comb of intermodulation products of the two drive tones[15]. In the time domain this frequency comb corresponds to a rapid oscillation with a slowly modulated amplitude and phase. Extracting the modulation phase allows us to resolve two Fourier coefficients of force, one which is in phase with the rapidly oscillating motion and its quadrature. These two components can be plotted as functions of the slowly varying amplitude $A$. Thus, the amplitude-dependent dynamic force quadrature $F_I(A)$ is the integrated conservative force in phase with the cantilever motion, and $F_Q(A)$ the dissipative force, in phase with the velocity[17] (see equations (5) and (6)). The transition from stick-slip to free-sliding dynamics of the AFM tip is revealed by a characteristic shape of these two force quadrature curves.

Figure 2a,b shows the measured force quadrature curves for a graphite surface at different interaction strengths, realized in the experiment by lowering the scanning feedback set-point, which moves the AFM probe closer to the surface. At each interaction strength the double curves show measurement with increasing and decreasing amplitude. The net interaction which loads the frictional contact is the sum of the adhesive forces and the cantilever bending force. The latter could in principle be measured by monitoring the vertical deflection of the cantilever. However, with the rather stiff cantilever used in this experiment we could barely resolve a change in static bending. With a softer cantilever adhesive forces cause a 'jump-to-contact' instability, making it very difficult to continuously regulate the load force. In our experiment we are able to smoothly regulate the load to observe a gradual evolution of the force quadrature curves, from zero interaction to sufficiently large interaction, where linear $F_I(A)$ is observed below a critical amplitude.

From simulations (see Theory section) we understand that this low-amplitude linear dependence of $F_I(A)$ corresponds to the tip apex being stuck to the surface. The measured cantilever motion is the result of elastic tip deformation. Above the critical amplitude where $F_I(A)$ has a distinct minimum, stick-slip dynamics begins. With increasing amplitude one observes a transition to smooth sliding, characterized by decreasing $F_I(A)$ and asymptotic approach of $F_Q(A)$ to a constant value. One can see how reducing the interaction force results in the gradual disappearance of the low-amplitude sticking regime. The horizontal scale of Fig. 2a,b also shows the maximum velocity of the tip base relative to the surface, $v_{max} = 2\pi f_0 A$, occurring when the cantilever crosses its torsional equilibrium point, twice each single oscillation cycle.

**Theory.** Our interpretation of the measured force quadrature curves in terms of stick-slip dynamics of a damped elastic asperity

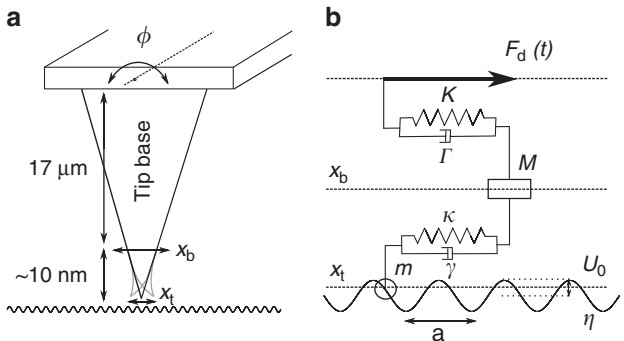

**Figure 1 | A schematic of the experiment and model, not to scale. (a)** The AFM cantilever undergoes a twisting oscillation at the resonance frequency of a high-Q torsional eigenmode. The resulting lateral motion of the tip base $x_b$ is dampened by frictional forces acting on the tip apex, $x_t$. **(b)** Schematic of the modified Prandtl–Tomlinson (PT) model used to describe the dynamical system. A driven support (cantilever base) is coupled to the nonlinear surface potential via a linear oscillator (torsional resonance) and elastic asperity (tip).

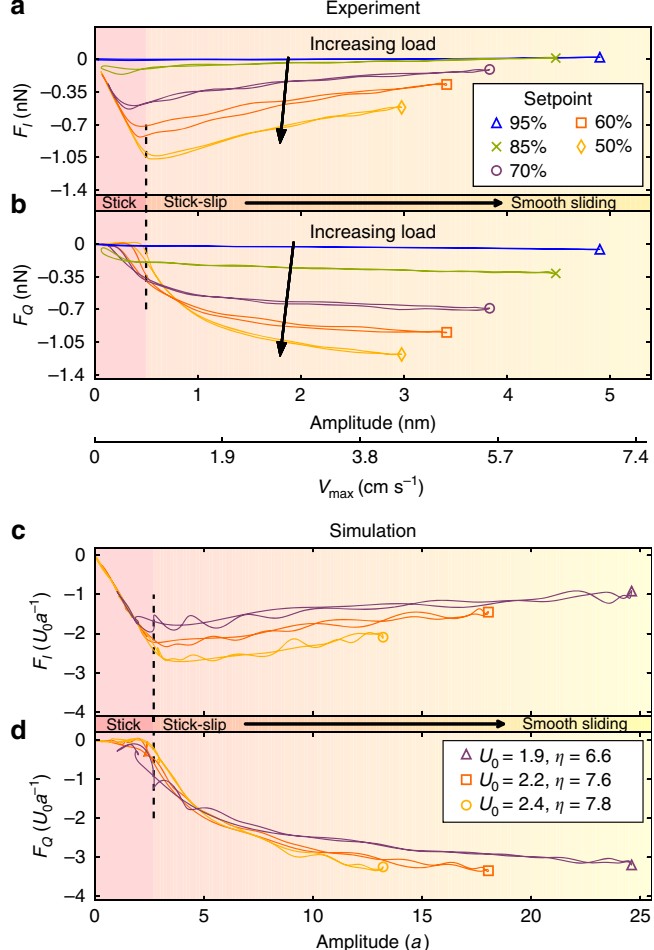

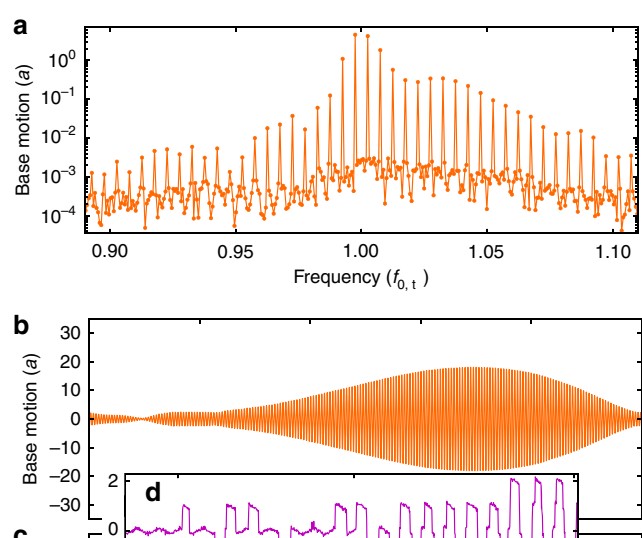

**Figure 2 | Force quadratures.** (**a,b**) Experimental curves at different probe heights showing continuous evolution from zero to finite load force. The load force is controlled by changing the feedback setpoint, given in the inset as % of free oscillation amplitude at the first drive frequency. With increasing load force (lower set-point) the low-amplitude motion of the cantilever occurs with the tip apex stuck to the surface and the slope of $F_I(A)$ gives the tip stiffness. At higher amplitude stick-slip behaviour gives way to smooth sliding, with $F_I$ decreasing toward zero and $F_Q$ approaching a constant value. Qualitatively similar behaviour is seen in the simulated force quadrature curves (**c,d**), derived from numerical integration of a modified Prandtl–Tomlinson model.

**Figure 3 | Simulated response.** (**a**) Simulated frequency domain response of the tip base $x_b$. The discrete comb of response at intermodulation frequencies $f_{IMP} = n_1 f_1 + n_2 f_2$ results from the periodic drive and the nonlinearity. (**b,c**) One period of steady-state motion in the time domain, for both tip base $x_b$ and the tip apex $x_t$ when $U_0 = 2.2$, $\eta = 7.6$ (orange curve in Fig. 2). The elastic tip allows for motion of the base even when the apex is stuck to the surface. The zoom inset (**d**) shows the stick-slip region.

At low drive amplitude the tip apex becomes stuck in a local minimum of the potential. The tip base continues to oscillate because the elastic tip can deform. With increasing drive amplitude the tip apex begins to slip between local minima of the potential as shown in Fig. 3d. When the drive amplitude is large enough, well-separated slips events give way to smooth-sliding over many minima in the surface potential.

## Discussion

The experimental curves in Fig. 2a,b show how the transition from stick-slip to smooth-sliding changes with applied load. With sufficient interaction strength, the tip can stick to the surface and the low amplitude slope of $F_I(A)$ gives the elastic stiffness of the tip $k$ (see Methods). For this probe we measure $k = 4 \text{ N m}^{-1}$, consistent with estimates made by other groups on similar probes[18,20]. At lower load force a detailed examination of the experimental curves shows hysteresis in the force quadratures, as the low amplitude sticking regime ($F_Q = 0$) gradually disappears with reducing load force. The simulations in Fig. 2c,d capture the qualitative shape of the force quadrature curves at higher load force, but at lower load force we find that the simulations become unstable, when the tip is just grazing the surface.

Intermodulation frictional force microscopy (ImFFM) provides a unique ability to quantitatively probe friction at high velocity with high spatial resolution. Only 2 ms are needed to measure the force quadrature curves at the nN force scale and cm s$^{-1}$ velocity scale. This time is short enough to scan at a typical rate for dynamic AFM (1 line per second, 256 pixels per line) and create an image of the transition from stick-slip to smooth sliding.

is based on comparison of the measured data with numerical simulation of a modified Prandtl–Tomlinson (PT) model[1,18–22]. In our model (see Fig. 1b) the particle is coupled via a spring and damper (damped elastic tip apex) to an intermediate support (rigid base of the tip), which in turn is coupled via a linear oscillator (cantilever torsional resonance) to a driven support (cantilever base). The inclusion of a damped elastic tip was necessary to explain the experimental data.

Figure 2c,d shows the simulated force quadratures (see Methods). Adjusting the parameters of the asperity, we can achieve good qualitative agreement between the experimental and simulated curves. Simulation allows for detailed examination of the system dynamics during the transition from stick-slip to sliding friction. In the frequency domain (Fig. 3a), the periodic motion of the tip base is represented by a frequency comb. In the time domain (Fig. 3b,c), the motion of both the tip base and tip apex is plotted over exactly one period $T = 1/\Delta f$, where $\Delta f = f_2 - f_1$ is the frequency difference of the two drive tones.

Figure 4a shows such a scan over a graphite surface, where the response amplitude at drive frequency $f_1$ is shown by colour. The feedback adjusts the probe height to keep this amplitude constant, and the feedback set point was changed at regular intervals during the scan, resulting in the horizontal bands seen in the image. Stable imaging was observed and there was no discernible evidence that the tip was damaging the surface, even at the highest load force.

Graphite serves as a well-studied test sample for demonstration of ImFFM but the image is basically featureless because the friction is so homogeneous. However, a change in the response is observed when scanning across an atomic step, seen as a diagonal feature in Fig. 4a. The inset Fig. 4b shows a zoom of the step region where the colour map codes for the critical oscillation amplitude at which $F_I(A)$ is minimum. In this region three pixels are marked, and the $F_I$ and $F_Q$ curves are shown in Fig. 4c,d with corresponding colour. Taking this critical amplitude for the onset

of sliding friction, one can see how the presence of the atomic step pushes the critical amplitude to larger values. The shape of the force quadrature curves near this step also differ considerably from those of the simulation of a corrugated surface, which did not include a step. We expect the presence of a step would inhibit smooth sliding, qualitatively explaining the broad minimum observed in $F_I(A)$ at much larger amplitude.

The zoom inset Fig. 4b is derived from the intermodulation spectra at each pixel and it shows features that are not present in any single amplitude or phase image. The zoom demonstrates the remarkable detail with which high velocity friction can be studied using ImFFM. The fact that neighbouring pixels (independent measurements) show similar critical amplitude demonstrates the extreme sensitivity of the method to small variations in frictional force, with spatial resolution limited only by the extent of the lateral tip oscillation, $2A \simeq 7.2$ nm for this scan. With its high spatial resolution, and its ability to capture the full amplitude dependence of friction at each image point, we anticipate that ImFFM will have large impact on our understanding of the origins of friction on heterogeneous nano-structured surfaces.

## Methods

**Sample, cantilever and calibration.** We scanned a freshly cleaved highly oriented pyrolytic graphite sample under ambient conditions. The cantilever (MPP-13120 also known as Tap525, Bruker) was calibrated using the noninvasive thermal noise method. The normal Sader method[23] is used to get the flexural stiffness $k_f = 53$ N m$^{-1}$ from the resonance frequency $f_{0,f} = 470$ kHz and quality factor $Q_f = 384$ determined by fitting the thermal noise spectrum of the first flexural eigenmode[24]. Similarly, the first torsional resonance $f_{0,t} = 2,400$ kHz, $Q_t = 704$ and the torsional Sader method[23] gives a torsional stiffness $k_\phi = 239 \times 10^{-9}$ N m rad$^{-1}$. The Sader method together with the fluctuation-dissipation theorem gives us the detectors inverse responsivity[25] $\alpha_t^{-1} = 1.2 \times 10^3$ rad V$^{-1}$. The torsional stiffness corresponds to a stiffness for in-plane forces acting on the tip, $K = k_\phi h_{tip}^2 = 827$ N m$^{-1}$ (manufacturer-specified tip height $h_{tip} = 17$ μm). We formulate the equations of motion below in terms of this equivalent lateral stiffness of the torsional eigenmode, with its associated mass $M = K(2\pi f_{0,t})^{-2}$ and damping coefficient $M\Gamma = K(2\pi f_{0,t} Q_t)^{-1}$, where $\Gamma$ is the width of the resonance.

**Force sensitivity and image resolution.** The sensitivity of a cantilever as transducer of force is enhanced by a factor $Q$ on resonance, in comparison with the quasi-static (zero frequency) limit. Owing to this enhancement the thermal

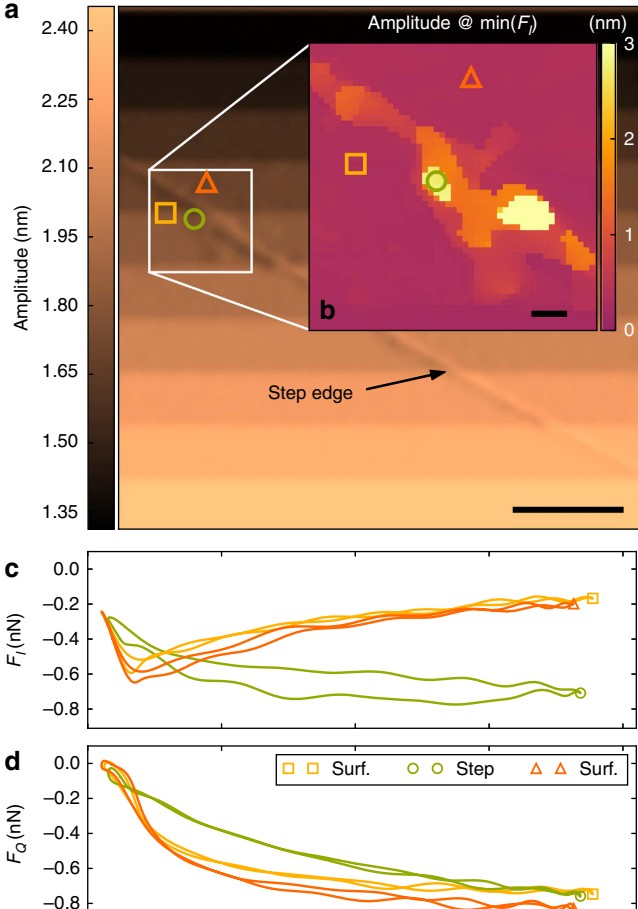

**a** Amplitude @ min($F_I$) (nm)

Step edge

**b**

**c**

**d**

□ □ Surf.    ○ ○ Step    △ △ Surf.

Amplitude (nm)

$V_{max}$ (cm s$^{-1}$)

**Figure 4 | Friction images of a highly oriented pyrolytic graphite surface.**
(**a**) The response amplitude at drive frequency $f_1$, used for scanning feedback. The horizontal bands are due to changes in the feedback set-point during the scan, where lower amplitude (darker) corresponds to the cantilever working closer to the surface. The diagonal feature is an atomic step. Scalebar 200 nm. (**b**) A zoom of the step region, where the image colour codes for the critical amplitude at which $F_I(A)$ is minimum. Scalebar 20 nm. (**c,d**) The force quadrature curves at the three pixels are marked with the corresponding colour and marker as in the image scans.

**Table 1 | Fixed parameters used for simulation of the PT model.**

| Symbol | Expression | Value | Description |
|---|---|---|---|
| $M$ | — | 2.5 | Tip mass |
| $K$ | — | 3.6 | Tip spring |
| $\gamma$ | — | 12 | Tip damping |
| $K$ | — | 40 | Cantilever spring |
| $f_0$ | $\sqrt{K/M}/2\pi$ | 0.001 | Cantilever resonance |
| $Q$ | $2\pi f_0/\Gamma$ | 500 | Cantilever quality factor |
| $\Delta f$ | $f_0/200.5$ | — | Frequency spacing |
| $f_1$ | $200\Delta f$ | — | First drive frequency |
| $f_2$ | $201\Delta f$ | — | Second drive frequency |
| $A_1$ | — | 0.21 | First drive amplitude |
| $A_2$ | — | 0.21 | Second drive amplitude |
| $a_0$ | — | 1 | Surface periodicity |
| $\sigma_{noise}$ | — | 6.1 | Strength of noise |

**Table 2 | Adjustable parameters.**

| $U_0$ | $\eta$ | Symbol in Fig. 2b | Description |
|---|---|---|---|
| 2.40 | 7.81 | Circle | Strongest interaction |
| 2.16 | 7.56 | Square | ↓ |
| 1.92 | 6.55 | Triangle | Weakest interaction |

Brownian motion of the cantilever can often be observed as a noise peak at resonance, where the Brownian motion noise exceeds the detector noise. In this case the minimum detectable lateral force acting on the tip is given by the thermal noise force, with power spectral density,

$$S_{FF} = 2k_B TM\Gamma = 2k_B T \frac{k_\phi^2}{h_{tip}^2 2\pi f_0 Q}. \tag{1}$$

Note that this noise force depends on the damping coefficient, not the stiffness, but it is convenient to express it in terms of stiffness, quality factor and resonant frequency, as the latter two quantities are easily accessible in the experiment. For a specified measurement bandwidth $B$ (inverse of the measurement time), the minimum detectable force is the force signal which just equals this noise $F_{min} = \sqrt{S_{FF}B}$. At the first torsional eigenmode of our cantilever with $B = 1$ kHz, we find $F_{min} = 0.88$ pN.

We compare with the quasi-static sensitivity where the measurement bandwidth is centered at zero frequency. Detector noise is typically limiting sensitivity with a noise equivalent force given by,

$$S_{FF}^{equiv} = \frac{S_{VV} k_\phi^2}{\alpha_t^2 h_{tip}^2}. \tag{2}$$

We take voltage noise $S_{VV} = 8.0 \times 10^{-12}$ V$^2$ Hz$^{-1}$ and inverse responsivity $\alpha_t^{-1} = 1.2 \times 10^{-3}$ rad V$^{-1}$ typical of our detector. Quasi-static measurements typically use a softer cantilever[6] $k_\phi \sim 3 \times 10^{-9}$ N m rad$^{-1}$ which, for the same $h_{tip} = 17$ μm and $B = 1$ kHz, gives $F_{min}^{equiv} = 13$ pN, a factor of 15 less sensitive than our experiment.

For quasi-static force measurement the time $B^{-1}$ and constant sliding velocity $v$ determine the distance over which the force is measured, which defines a minimum feature size $\delta = vB^{-1}$. Increasing the measurement bandwidth (decreasing the measurement time) improves resolution, but at the expense of force sensitivity. With dynamic force measurement the minimum feature size is independent of the measurement bandwidth, given only by the amplitude of sliding oscillation $\delta = 2A$, or in terms of the maximum velocity achieved in the oscillation $\delta = v_{max}(\pi f_0)^{-1}$. High resolution (small $\delta$), high force sensitivity (small $F_{min}$) and high velocity (large $v_{max}$) are all achieved with a small bandwidth measurement on resonance using a cantilever with large $f_0$ and large $Q$.

**Intermodulation measurement and scanning feedback.** The cantilever is excited with a split-piezo actuator at two frequencies $f_1$, $f_2$ centered on torsional resonance $f_{0,t}$ and separated by $\Delta f = f_2 - f_1 \ll f_{0,t}$. The drive frequencies $f_1$ and $f_2$ are chosen such that they are both integer multiples of $\Delta f$. The drive is synthesized, and the response is measured with a synchronous multifrequency lockin amplifier (Intermodulation Products AB; http://www.intermodulation-products.com/)[26] which also calculates the feedback error signal used by the host AFM. A proportional-integral feedback loop adjusts the probe height to keep the $f_1$ response amplitude at the set-point value. The exact type of feedback used is not critical to the method, only that it is responsive enough to track the surface topography at the desired scan speed. We also desire that the feedback error is small enough, such that we can approximate the probe height as being constant during the time $T = (\Delta f)^{-1}$ needed to measure the response. This time defines one pixel of the 42 amplitude and phase image-pairs acquired at each frequency, during a single scan.

**Model and equations of motion.** A schematic representation of the model can be seen in Fig. 1. Performing force balance on both masses results in two coupled one-dimensional equations of motion in the lateral position of the tip apex $x_t$, and tip base $x_b$.

$$M\ddot{x}_b = -Kx_b - \Gamma M\dot{x}_b + F_c(d, \dot{d}) + F_d(t), \tag{3}$$

$$m\ddot{x}_t = -F_c(d, \dot{d}) - F_{surf}(x_t, \dot{x}_t). \tag{4}$$

The coupling force $F_c = kd + m\gamma\dot{d} + F_{noise}$ is linear in the deformation of the tip, $d = x_t - x_b$, and damping linear in $\dot{d}$. $F_{noise}(t)$ is a random noise force with a Gaussian distribution[1,27]. The strength of the noise is characterized by the standard deviation $\sigma_{noise}$, given in Table 1. The nonlinear frictional force $F_{surf} = -\eta\dot{x}_t - \frac{\partial}{\partial x_t}U(x_t)$ is derived from damped motion in a periodic potential $U(x_t) = U_0 \cos(2\pi x_t/a_0)$. The drive force $F_d = K[A_1 \cos(2\pi f_1 t) + A_2 \cos(2\pi f_2 t)]$ is applied at two frequencies as described above.

**Dynamic force quadratures.** We probe friction by measuring two dynamic quadratures of the nonlinear force which is perturbing the harmonic motion of the torsional resonance. The method was originally developed for normal forces and flexural resonance by Platz et al.[17,28] From the measured intermodulation spectrum and the calibrated transfer function of the torsional eigenmode, we determine the oscillation amplitude dependence of the force quadratures, without any assumptions as to the nature of the perturbing force. For the model described above, $F_I$ gives the integrated coupling force $F_c$ that is in phase with the motion

of the tip base, and $F_Q$ that is quadrature to the motion, or in phase with the velocity.

$$F_I(A) = \frac{1}{T} \int_0^T F_c(x_b, \dot{x}_b)\cos(\omega_0 t)dt, \tag{5}$$

$$F_Q(A) = \frac{1}{T} \int_0^T F_c(x_b, \dot{x}_b)\sin(\omega_0 t)dt, \tag{6}$$

where

$$x_b(t) = A\cos(\omega_0 t). \tag{7}$$

When $F_{fric} \gg F_c$, the tip apex is stuck in a minimum of the surface potential, $x_t \approx$ const, and motion of the tip base is due to tip deformation alone. In this case we can solve the integrals in equations (5) and (6),

$$F_I(A) = -\frac{kA}{2} \text{ and } F_Q = -\frac{m\gamma v_{max}}{2}. \tag{8}$$

Thus, the slope of $F_I(A)$ at low amplitude and high load gives the stiffness of the asperity. Similarly, the slope of $F_Q(A)$ gives the damping of the asperity, which is not resolvable in our experiment.

**Simulation.** We simulate the experiment by numerical integration of the model equations (3) and (4) using CVODE[29]. The dynamical system is converted to four first-order differential equations, characterized by two resonant frequencies: $\omega_{0,b} = \sqrt{K/M}$ and $\omega_{0,t} = \sqrt{k/m}$. When $\omega_{0,t} \gg \omega_{0,b}$ the adaptive time-step integrator becomes rather slow. We chose $\omega_0^t/\omega_0^b \sim 300$, which is at least one order of magnitude smaller than experiments, but still large enough to simulate the dynamics qualitatively so that we can explore the parameter space of the model in a reasonable time (each simulation takes 200 s on an Intel Core i7, 3.50 GHz PC). We simulated with normalized values where (length) = 1.42 Å, (mass) = $4.78 \times 10^{-25}$ kg and (time) = $4.46 \times 10^{-13}$ s. The parameters are given in Tables 1 and 2. To simulate different interaction strengths, we vary the surface potential $U_0$ and dissipation of the tip $\gamma$ as in Table 2. Our choice of simulation parameters means that the simulated frequency of surface-induced force pulses on the tip $f_{surf} \sim (A/a_0)f_{0,t}$ is about an order of magnitude smaller than in the experiment. Nevertheless, our simulation is able to capture the qualitative shape of the force quadrature curves at high velocity and high interaction strength. However, with these simulation parameters we are not able to reproduce the experiment at low velocity and low interaction.

**Data and code availability.** The data, analysis and simulation code that support the findings of this study are available from the corresponding author PAT upon reasonable request.

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

## Acknowledgements

We gratefully acknowledge financial support from the Swedish Research Council (VR), the Knut and Alice Wallenberg Foundation and the Olle Engkvist Foundation. We also acknowledgement the use of methods and analysis code originally developed by Daniel Platz, as well as fruitful discussions with Mark Rutland and Roland Bennewitz.

## Author contributions

All authors contributed to discussion and interpretation of the experimental data, model and simulations. P.-A.T. did the measurements and data analysis, performed the simulations and generated all figures. A.S.d.W. contributed with model development and simulation code. R.B. and D.F. contributed to the experiments and simulation code. D.B.H., A.S.d.W. and P.-A.T. contributed to the writing of the manuscript.

## Additional information

**Competing financial interests:** D.B.H. and D.F. are part owners of the company Intermodulation Products AB, which manufactures and sells the multifrequency AFM add-on system used in this work.

**Publisher's note**: 

