## [Peer Review File · Nature Communications]

Reviewers' comments:

Reviewer #1 (Remarks to the Author):

This paper introduces a new method that can measure the stick slip friction at a scanning speed of several cm/s. This method enables researchers to explore more about the stick slip. Furthermore, the speed condition in experiment can better match that in simulations, which is generally difficult. The paper may be worthy of consideration for publication. However, more detailed explanations to some of the concepts in this paper are needed to address the concerns below.

Major concerns:

1. The authors should explain more about the conservative force and dissipative force. It is not clear how the authors relate these forces with stick-slip friction.
2. The authors should explain how they controlled or detected the amplitude and frequency variable to plot the amplitude-force relation.
3. Have the authors used the method at a relatively low speed and compared it to the results from the quasi-static method?
4. Regarding the accessible velocity range: in Fig. 2, the linear regimes of conservative forces indicate that the tip sticks to the substrate, which means there is a minimum dynamic speed at each load. According to this figure, this minimum speed is about 0.5 cm/s. This is quite fast. The authors can probably reduce this minimum by further reducing the speed. However, this still means this technique can NOT access the full velocity range (zero to cm/s) at all loads as author claimed. The authors should discuss this and make a plot of minimum speed verse load.
5. Regarding direct evidence of stick-slip: can the author provide some direct evidence of the stick-slip motion?
6. What is the value of each load in Fig. 2? Can the authors also show two lines with 0 load and without contact on the same plot?
7. Could the authors provide some TEM or blind tip-reconstruction images of the tip before and after experiments? It is important to see if tip undergoes any shape change during the experiments (especially after high loading).
8. Could the authors compare and discuss the parameters between experiments and model? There are quite a few parameters in the simulations. A table would help.

[1] Liu, Xin-Z., et al. "Dynamics of atomic stick-slip friction examined with atomic force microscopy and atomistic simulations at overlapping speeds." *Physical review letters* 114.14

(2015): 146102.

Minor concerns:

1. Why do the authors take into account the damped elastic deformation when using the PT model?
2. In the 2nd paragraph the authors state "with this quasi-static method stick-slip behavior can be observed, but only up to velocities ~ 10 nm/s", However there was a recent paper [1] in which the fastest speed can reach ~ 580 $\mu\text{m/s}$ with a modified sample holder using quasi-static friction. The author should refer to and consider this paper.
3. The maximum amplitude (also speed) changes with load. Can the authors make a plot of maximum speed versus load?
4. The lines from different load or surface/steps in figure 1 and 3 are very hard to distinguish, especially when printed in black and white. I would suggest using different styles of lines (dash, dot) for different conditions.

Reviewer #2 (Remarks to the Author):

In my view this is a highly relevant paper on a new method to characterize the velocity dependence of friction on the nanoscale. The paper is well suited for Nature communication after major revision. In particular the theoretical model needs to be revised.

In detail:

The authors report on a novel method to measure velocity dependent friction coefficients on the nanometer scale with an atomic force microscope. The basic idea is to analyze nearly resonant torsional oscillations of the cantilever when the tip is in contact with the sample. Unlike other methods (Refs 10-12) the cantilever is not driven directly at resonance but at two frequencies slightly above/below. Due to mixing of the signals from both oscillations intermodulation signals occur. These intermodulation signals can be used to reconstruct frictional forces (more precisely: force quadratures of the elastic and dissipative lateral forces). The scan velocity can be varied by varying the oscillatory amplitude. Due to the high torsional frequency, maximum velocities of up to 5cm/s could be achieved.

On HOPG a stick-slip regime could be identified for velocities up to 0.5 cm/s (load dependent). At higher velocities smooth sliding occurred. Such a behavior is typical for friction measured on HOPG. The paper is well presented and the arguments and the interpretation of data are well developed (maybe except for the comparison theory/experiment).

The specific originality of the paper is the elegant approach to overcome current "speed limits" in

AFM. Currently it is very difficult to carry out high speed friction measurements with an atomic force microscope. The reason is rather technical: Although an increasing number of designs for high-speed AFM have been developed (Ando, Hansma, Schitter and others) it is very difficult to limit the instrument noise in the feedback circuit. Also thermal noise plays an important role in measuring atomic friction with these high speed AFMs because rather soft cantilevers have to be used. The benefit of the method proposed by the authors is that (1) the cantilever is stiff and (2) the scan amplitudes are very small such the friction coefficient can be measured locally.

In general, the experimental approach is well justified, because the intermodulation method is well established in other contexts. Thus experimental data are highly reliable. Data are well presented and relevant plots are shown and discussed.

Thus, I am convinced that the method will have high impact in the nanotribology community. In my view, however, the discussion, i.e. the comparison between experiment and simulation needs to be strengthened:

(1) Although the experiments were carried out at room temperature, instrumental noise might play a role and help to develop a more realistic model. (see e.g. Yalin Dong, Hongyu Gao, Ashlie Martini, and Philip Egberts Phys. Rev. E 90, 2014, 012125).

(2) In addition, the authors need to address the fact that at low velocities sliding on HOPG is a thermally activated process. Thermal activation in the PT model is well established for this regime so it is obvious that a model without thermal noise cannot describe the low velocity regime correctly. The statement "This discrepancy is likely due to the absence of the thermal noise" is certainly correct but the authors need to discuss why their model still is useful. More precisely: Why can thermal activation be neglected under certain operating conditions? Here papers by E. Riedo, E. Gnecco, E. Mayer and others might be helpful to develop convincing arguments.

(3) If possible, the paper would be strengthened a lot if thermal activation could be included into the simulation (see e.g. Martin H. Muser, Phys. Rev. B 84, 125419, 2011).

(4) The authors should be aware that stiff cantilevers increase the risk that graphene flakes may be cut out of the graphite by the tip such that graphite-graphite interaction is measured. (In STM moiré patterns can be observed). In friction experiments then superlubricity may occur (Yilun Liu, François Grey & Quanshui Zheng, Scientific Reports 4, 4875 (2014) doi:10.1038/srep04875). To rule out such an effect it would be helpful to know how reproducible the results were with different AFM tips and what the tips looked like after the experiment (SEM).

We thank both reviewers for their careful reading of the manuscript and their helpful comments. Below we reply to all comments point-by-point.

Reviewer #1:

Major concerns:

1. The authors should explain more about the conservative force and dissipative force. It is not clear how the authors relate these forces with stick-slip friction.

We have attempted to clarify the nature of the dynamic force quadratures $F_I(A)$ and $F_Q(A)$ by re-writing the introductory paragraph of the Experimental section. We describe their role in revealing the transition from stick-slip to free-sliding friction. More detailed discussion and the appropriate equations were given in the Methods section. We also changed one reference to our previous work, to include a paper which explains the mathematics in more detail. Our objective with this paper is to describe the force quadratures in a physically intuitive way, without dragging the reader through a lot of detailed math.

2. The authors should explain how they controlled or detected the amplitude and frequency variable to plot the amplitude-force relation.

We changed the introductory paragraph of the Experimental section to address this point. In essence, the multifrequency response realizes a slow modulation of the amplitude and phase. Our technique allows us to track the modulation of both the amplitude and the phase, by coherent measurement (lockin with one reference oscillation) of multiple frequency components of the motion.

3. Have the authors used the method at a relatively low speed and compared it to the results from the quasi-static method?

We did not use the quasi-static method with this cantilever. The low-velocity (quasi static) force sensitivity of our stiff torsional cantilever not very high. Thus, we would have to apply very large load force to get a measurable static twisting of the cantilever. Such large load force would probably destroy the tip and sample. We discuss this point in the methods section, and in response to other comments below.

4. Regarding the accessible velocity range: in Fig. 2, the linear regimes of conservative forces indicate that the tip sticks to the substrate, which means there is a minimum dynamic speed at each load. According to this figure, this minimum speed is about 0.5 cm/s. This is quite fast. The authors can probably reduce this minimum by further reducing the speed. However, this still means this technique can NOT access the full velocity range (zero to cm/s) at all loads as author claimed.

We believe that there is a point of confusion here. The velocity (which is constant for traditional AFM friction methods) should not be compared with our v_{max} , which is the maximum speed during a single cycle of the torsional oscillation at frequency 2.4MHz. The cantilever velocity goes to zero at each turning point of every single oscillation cycle. The tip can undergo stick, to stick-slip, to sliding motion all in one single oscillation cycle, depending on the amplitude of that individual oscillation cycle. The minimum of $F_I(A)$ happens when the forcing amplitude (or cantilever motion amplitude) is large enough to overcome the sticking barrier, or static friction of the surface. We have added a few sentences to the last paragraph of the Experimental section to

clarify this point.

The authors should discuss this and make a plot of minimum speed verse load.

With the above arguments, we can not uniquely define a velocity at which this transition occurs (only a maximum velocity of the oscillation cycle).

5. Regarding direct evidence of stick-slip: can the author provide some direct evidence of the stick-slip motion?

The stick-slip events at high velocity result in force pulses with frequency content much higher than the resonant frequency of the cantilever. Therefore, the cantilever can not respond to these force pulses and we can not measure the force pulses directly. However, the beauty of the intermodulation technique is that it allows us to use the nonlinearity of the interaction itself, to 'down-convert' (i.e. mix to lower frequency) this high frequency dynamics to the frequency band near resonance, where the high sensitivity of the cantilever can read the response. The agreement between our simulations and experiment verifies this picture. In the simulation, we can resolve the individual stick-slip events (see figure 3).

Thus, the referee is correct in saying that we do not 'directly' resolve individual stick-slip events. No method can detect these events at such high velocity. But, due to the high force sensitivity of our dynamic method, and the clever use of intermodulation (frequency mixing) we do see a clear signature of these events in the shape of the $F_I(A)$ and $F_Q(A)$ curves.

6. What is the value of each load in Fig. 2? Can the authors also show two lines with 0 load and without contact on the same plot?

We can not directly measure the load force, which should include both the cantilever force, and the adhesion force. One could in principle measure cantilever vertical deflection, which would give the cantilever force, but for the cantilever used in this experiment, which was rather stiff, we could only resolve very small changes in static bending. Note that the feedback setpoint is controlling the load force. Because the feedback works with a narrowband measurement of the torsional resonance, it is extremely sensitive to changes in interaction with the surface. Hence, we have very fine control over the net load force, but we can not resolve exactly what the net load is, because we do not know the exact form of the adhesive forces, and we can not resolve a static bending of the beam. We have re-written the second paragraph of the Experiment section to address these points.

We do measure the free motion (without contact) where the tip does not interact with the surface. In this case $F_I(A)$ and $F_Q(A)$ are zero by definition. Of course, we verify that this is indeed the case when we measure. We have added additional curves to the FI and FQ experimental plots of fig. 2, which demonstrate that we see a continuous evolution of these curves from zero interaction. We have modified the caption of figure 2, and added the set-point values.

7. Could the authors provide some TEM or blind tip-reconstruction images of the tip before and after experiments? It is important to see if tip undergoes any shape change during the experiments (especially after high loading).

We did not make a detailed study involving TEM. We agree that such a study would be interesting and important future work. However, figure 4 does show that the tip is able to make high resolution images of an atomic step on graphite. The inset has size bar 20nm. Therefore the tip can not be too damaged. Note that each pixel in this image is one measurement of the FI and FQ curves. After scanning many pixels at increasing loads (12800 measurements at each load

force), we do not see any discernible change in the step resolution, nor do we see any change in the measured FI(A) and FQ(A) curves. This is not surprising because the load forces are small, and very well regulated, due to the extreme force sensitivity of the torsional resonance, as we discuss in the introduction (detailed in Methods). We have added a sentence to the second paragraph of the discussion section to clarify this point.

8. Could the authors compare and discuss the parameters between experiments and model? There are quite a few parameters in the simulations. A table would help.

It is convenient to use normalized parameters with this phenomenological model. We put a short discussion of this in the Methods section. We have presented this in a table format to make it more accessible. Thank you for this suggestion.

Minor concerns:

1. Why do the authors take into account the damped elastic deformation when using the PT model?

This was necessary to get the simulations and experiment to match for large velocities. We needed to add the damping of an elastic tip, to explain the experimental data. It appears that in our experiments the majority of the energy is dissipated in the tip, not in the substrate. We added a sentence at the end of the first paragraph in the Theory section to make this point more clear.

2. In the 2nd paragraph the authors state "with this quasi-static method stick-slip behavior can be observed, but only up to velocities ~ 10 nm/s", However there was a recent paper [1] in which the fastest speed can reach ~ 580 $\mu\text{m/s}$ with a modified sample holder using quasi-static friction. The author should refer to and consider this paper.

We thank the referee for pointing this out. We have added the reference.

3. The maximum amplitude (also speed) changes with load. Can the authors make a plot of maximum speed verse load?

The load is controlled by feedback-loop, which keeps the amplitude at a certain fraction of the free amplitude (set-point). Because we do not independently measure the load, we can not make such a plot (see point 6 above).

4. The lines from different load or surface/steps in figure 1 and 3 are very hard to distinguish, especially when printed in black and white. I would suggest using different styles of lines (dash, dot) for different conditions.

Thank you for the comment. We have updated the figures.

Reviewer #2:

(1) Although the experiments were carried out at room temperature, instrumental noise might play a role and help to develop a more realistic model. (see e.g. Yalin Dong, Hongyu Gao, Ashlie Martini, and Philip Egberts Phys. Rev. E 90, 2014, 012125).

We agree with the referee here, that including noise might help to better explain the data. In the experiment there is certainly a thermal noise force driving the cantilever (we measure this when we calibrate), and there may also be additional 'instrumental' noise (e.g. voltage noise in the signal driving the torsional actuator). Both sources of noise correspond to a random force acting on the system. We have now included such a random force into our simulations (see updated Methods-section). With this noise included we see a slightly better qualitative agreement between experiment and theory in the low-velocity regime, but significant discrepancy between experiment and our simplified model remain at low load force.

(2) In addition, the authors need to address the fact that at low velocities sliding on HOPG is a thermally activated process. Thermal activation in the PT model is well established for this regime so it is obvious that a model without thermal noise cannot describe the low velocity regime correctly. The statement "This discrepancy is likely due to the absence of the thermal noise" is certainly correct but the authors need to discuss why their model still is useful. More precisely: Why can thermal activation be neglected under certain operating conditions? Here papers by E. Riedo, E. Gnecco, E.Mayer and others might be helpful to develop convincing arguments.

Please see response to concern (1). The problem is that our simulations are not really in the low velocity regime because the characteristic frequency of the cantilever dynamics, is not very much lower than that of the tip dynamics. We discussed this point the Simulation subsection of the Methods section.

Here we could improve our simulations with more brute force (i.e. more powerful computer), which would allow us to simulate in a more reasonable time, perhaps achieving numbers closer to that of the experiment. However, even in the experiment we reach such high velocity, that it is not clear if thermal equilibrium escape dynamics could explain the stick-slip events. Our objective with this paper was to introduce and explain this very new technique which requires that we simulate the entire cantilever dynamics, which is our focus here.

(3) If possible, the paper would be strengthened a lot if thermal activation could be included into the simulation (see e.g. Martin H. Muser, Phys. Rev. B 84, 125419, 2011).

We have now included noise in the simulations. Thank you for pointing this out.

(4) The authors should be aware that stiff cantilevers increase the risk that graphene flakes may be cut out of the graphite by the tip such that graphite-graphite interaction is measured. (In STM moiree patterns can be observed). In friction experiments then superlubricity may occur (Yilun Liu, François Grey & Quanshui Zheng, Scientific Reports 4, 4875 (2014) doi:10.1038/srep04875). To rule out such an effect it would be helpful to know how reproducible the results were with different AFM tips and what the tips looked like after the experiment (SEM).

We did not make a detailed study involving the state of the tip after experiments. We agree that such a study would be interesting and important future work. However, figure 4 does show that the tip is able to make high resolution images of an atomic step on graphite. The inset has size bar 20nm. Therefore the tip can not be too modified during the scan. Note that each pixel in this

image is one measurement of the FI and FQ curves. After scanning many pixels at increasing loads (12800 measurements at each load force), we do not see discernible change in the step resolution, nor do we see any change in the measured FI(A) and FQ(A) curves. This is not surprising because the load forces are small, and very well regulated, due to the extreme force sensitivity of the torsional resonance, as we discuss in the introduction (detailed in Methods). We have added a sentence to the second paragraph of the discussion section to clarify this point.

Reviewers' comments:

Reviewer #1 (Remarks to the Author):

The authors have addressed most of the concerns raised previously. The referee has the following remaining concerns that should be addressed before publication is warranted:

1. What limits the maximum velocity in figure 2? Is there a load at which the maximum load causes the range of accessible velocities to be much smaller than the cm/s range?
2. How do you explain the shape of the F_i curve in figure 4c, for the "step" case? It does not follow the trend of any of the curves in figure 2. It also has no clear minimum, though page 4 refers to the minimum for the onset of smooth sliding. Are the authors implying that there is no smooth sliding on the step and the entire measured F_i curve is in the "stuck" regime? What does that say about effective tip stiffness at the step?
3. It is still unclear how thermal would affect these measurements. The authors should explain more clearly the effect of the added noise term.
4. Is there a length associated with the stick slip motion? Could one use this technique to compare the lattice spacing of different materials or Moire patterns on layered materials? In other words, is this technique resolving atomic stick slip or the more irregular stick slip that is seen at larger length scales? It is unclear from the manuscript. Are there any approaches to calibrate the load forces and to resolve stick-slip experimentally?
5. I suggest the authors clarify "setpoint" in the manuscript, e.g., what does the percentage mean in Fig. 2(a) and (b)? It should correspond to the free amplitude. Right now it may be confused with the controlled amplitude. This can help readers understand how load varies with the set point as well as how the adhesive force changes.
6. Regarding the force quadratures plots, is it possible to mark different regimes (stick, stick-slip, and sliding motion) with shaded area on those plots? I know the "stick-slip" regime may be very small according to Fig. 3 c). Overall, it would help if the authors explained more clearly how the $F_I(A)$ and $F_Q(A)$ curves show stick-slip friction.
7. It is still unclear that how the authors relate the results with previous studies on velocity dependent friction. Can the authors compare the results with previous simulation results at a high velocity qualitatively? For example, the simulation results from Liu, Xin-Z., et al. Physical review letters 114.14 (2015): 146102 and other literature such as papers cited within that work.
8. Is it possible to change a relatively soft cantilever that allows measuring the bending and it is still available for this method?

Stylistic concerns:

There are still multiple spelling/grammar errors.

The second-to-last full sentence in the first column of page 2 is confusing and should be reworded. It is the sentence that begins, "At higher amplitude stick-slip dynamics..."

Reviewer #2 (Remarks to the Author):

In my view, the authors have convincingly replied to the referees' comments. I very much appreciate that a noise term has been included in the simulations.

Thus I recommend the paper for publication.

Reviewers' comments:

Reviewer #1 (Remarks to the Author):

The authors have addressed most of the concerns raised previously. The referee has the following remaining concerns that should be addressed before publication is warranted:

We thank the referee for these very helpful additional comments and for the second, careful reading of our manuscript. Considering the comments and questions below, we have made numerous changes to the manuscript. Most changes were simply a matter of rearranging the text to improve readability and clarity, with out adding new content. In some cases we added new content by expanding the discussion to address the questions and comments below. We hope that the changes we have made to the manuscript and our comments below will clarify the referees outstanding concerns.

1. What limits the maximum velocity in figure 2?

The upper limit of the v_{\max} is determined by two factors: the stored energy (or circulating power) in the torsional resonance, and the load force. The former we control by how hard we drive the resonance, and we characterized it with the measured maximum amplitude of the free oscillation. The latter is controlled by the setpoint. The setpoint is given as a percent of the free oscillation amplitude. We added discussion in the text to clarify this point (see below).

Is there a load at which the maximum load causes the range of accessible velocities to be much smaller than the cm/s range?

Yes. The maximum velocity (amplitude of harmonic motion) of the tip base can be suppressed continuously to zero. When we increase the load force (move closer to the surface) the maximum velocity (amplitude) drops, as seen in figure 2. If we move too close to the surface, the load will become so large that the maximum velocity (amplitude) will never pass the minimum of $F_I(A)$ and in this case the tip is always stuck to the surface, never making the transition to stick-slip motion.

Changes: We modified the DISCUSSION section to address these points.

2. How do you explain the shape of the F_I curve in figure 4c, for the "step" case? It does not follow the trend of any of the curves in figure 2. It also has no clear minimum, though page 4 refers to the minimum for the onset of smooth sliding.

The referee is correct that the curve does not have a 'clear' minimum. But there is in fact a minimum at larger amplitudes, and this shift to higher amplitudes is clearly dependant on the exact location of the probe in relation to the 'step'. The inset of figure 4 shows that neighboring pixels (independent measurements of $F_I(A)$), even on different scan lines, all show the same critical amplitude at which the minimum occurs. Clearly, the shape of the F_I curve is very sensitive to the precise spatial relation between the step and the oscillating probe.

To fully understand the shape of the F_I curve in the step region, we would need to simulate the dynamics in the presence of the step. Note that our simple Prandtl-Thomlison model does not include the step. However, qualitatively we expect that the step should cause an additional larger barrier at which the tip can get stuck. As shown in our simulations without a step, the onset of

stick-slip occurs beyond the critical amplitude where the minimum of $F_I(A)$ occurs. So, the fact that the presence of the step pushes this critical amplitude to higher values, is consistent with what we would qualitatively expect. We would also expect that the shape of $F_I(A)$ would change.

Are the authors implying that there is no smooth sliding on the step and the entire measured F_I curve is in the "stuck" regime? What does that say about effective tip stiffness at the step?

To the extent that this very broad minimum is pushed out above the maximum amplitude, then yes, there would be no transition to 'smooth' sliding. The tip may slide smoothly up to the step, then get stuck at the step, requiring extra energy (force) to get past the step. To fully describe this, we would need to model the step in the simulation. This could be done, but it is beside the main point of this paper, which is to introduce and validate a new way of measuring high-speed friction at the nanoscale.

Changes: We modified the DISCUSSION section to address these points.

3. It is still unclear how thermal would affect these measurements. The authors should explain more clearly the effect of the added noise term.

The measurement always has noise. Typically measurements of stick-slip are affected by detector noise. However, the noise that we are interested in is the random thermal equilibrium 'noise' force that is present due to the damping terms in our model (fluctuation dissipation theorem). There are in principle two sources of random force (two damping terms in our model), one acting on the cantilever body, and one acting on the tip.

At the request of reviewer 2, in the second version of the manuscript we added the tip noise force to the simulation. As we suspected, the noise did not qualitatively change the basic picture of the transition from stick, to stick-slip, to free-sliding friction in the $F_I(A)$ and $F_Q(A)$ curves. However, there is a quantitative change. The noise force gives the tip a little extra kick, which helps the tip escape from its stuck state, causing the transition to occur at slightly lower amplitude (max velocity) than would be the case without noise. However, the basic periodic motion of the two-frequency-driven nonlinear dynamical system, is unaffected by this additional random force because it is so very much smaller than the drive force.

Note that here our dynamic method differs considerably from the quasi-static methods, where one resolves individual slip events between minima in the surface potential. In the work of Liu *et al.* (reference suggested below) the frequency of these events reaches 30 kHz at the largest velocity that they were measured (not simulated). This frequency is much smaller than in our case, where we have at most some 50 events in one cycle of oscillation with period 0.5 microsec (i.e. event frequency reaching 100 MHz).

Because the time between each individual slip events is so very much smaller at high velocity, the random noise force will have a very much lower probability of being large enough to significantly contribute to 'pushing to tip over the top', or causing the slip to happen. Thus, faster measurement is less susceptible to noise.

Nevertheless, we do not want to go in to a detailed discussion of noise in this work. While fluctuations are indeed important to understanding friction, they are the subject of future work. To really say anything conclusive about force fluctuations we need to study both sources of force noise mentioned above, and we need to systematically study the temperature dependence of F_I and F_Q curves. This study will have to wait until we can build a variable temperature AFM.

Changes: We modified the introduction (paragraphs 2-4) to make more clear the difference between quasi-static and dynamic measurement. The answers to these questions are also somewhat contained in our more detailed discussion of force sensitivity, in the METHODS section.

4. Is there a length associated with the stick slip motion?

Yes, you could say that there is a length scale associated with the transition from sticking, to stick-slip. It is the amplitude of oscillation (of the tip base) at the minimum in $F_I(A)$, what we call the critical amplitude, for the onset of stick-slip motion. Note that this amplitude of base motion is not the same thing as the displacement of the tip itself, because the tip is not rigid.

Could one use this technique to compare the lattice spacing of different materials or Moire patterns on layered materials? In other words, is this technique resolving atomic stick slip or the more irregular stick slip that is seen at larger length scales?

We believe that the individual slip events between the tip and surface are indeed at the atomic scale. Note that in figure 2a, the transition between sticking and sliding occurs at a value around 4 Å (depending on the load force). This distance is only a bit larger than the lattice parameter of graphite (table value of about 2.5 Å). We see a somewhat large amplitude of motion of the tip base because the elastic tip stretches before the individual slip event occurs.

It is unclear from the manuscript. Are there any approaches to calibrate the load forces and to resolve stick-slip experimentally?

We can not independently measure the load force, mainly because we have rather strong adhesion forces which dominate the load force in our case. When working with low load force, as in our experiments, the adhesion force dominates the total load, and we can not approximate the net load as being simply proportional to the flexural bending of the cantilever. A softer cantilever would give larger static bending for some given force, allowing us to measure small changes in the cantilever bending force, but we still do not know the (in our case larger) adhesion force.

We have done experiments with softer cantilevers but they do not work due to the 'jump to contact' instability. The strong adhesion force pulls the cantilever to the surface, causing a large load to rapidly turn on, which kills the torsional oscillation. We need a stiff cantilever to work against these strong adhesion forces, so that we can regulate the load continuously from zero without the jump-to-contact instability.

Although we can not measure the load forces directly, we believe that they are in fact much smaller than those typically used in quasi-static friction measurements. As we point out in our introduction, the torsional resonance is much more sensitive to frictional forces than quasi-static measurement. Because of this enhanced sensitivity, we can see a large change in response at resonance, due to a very weak normal (load) force and the resulting weak frictional force.

Changes: We expanded on the discussion of jump-to-contact in paragraph two of the EXPERIMENT section. We rearranged the introduction (paragraphs 2-4) to make more clear the improved sensitivity of the dynamic method.

5. I suggest the authors clarify "setpoint" in the manuscript, e.g., what does the percentage mean in Fig. 2(a) and (b)? It should correspond to the free amplitude. Right now it may be confused with the controlled amplitude. This can help readers understand how load

varies with the set point as well as how the adhesive force changes.

We did this, both in the text and in the caption to figure 2.

6. Regarding the force quadratures plots, is it possible to mark different regimes (stick, stick-slip, and sliding motion) with shaded area on those plots? I know the "stick-slip" regime may be very small according to Fig. 3 c). Overall, it would help if the authors explained more clearly how the $F_I(A)$ and $F_Q(A)$ curves show stick-slip friction.

We modified figure 2, using a shaded color background and a color bar with text, to mark the various regions.

7. It is still unclear that how the authors relate the results with previous studies on velocity dependent friction. Can the authors compare the results with previous simulation results at a high velocity qualitatively? For example, the simulation results from Liu, Xin-Z., et al. Physical review letters 114.14 (2015): 146102 and other literature such as papers cited within that work.

We changed in introduction (paragraphs 2-4) to more clearly discuss the differences between the 'traditional' quasi-static force method where one studies individual stick-slip events at low velocity, and our method where we measure the integrated effect of many stick-slip events at very high velocity. We have added the paper of Liu *et al.* to our list of references.

8. Is it possible to change a relatively soft cantilever that allows measuring the bending and it is still available for this method?

See the discussion to point 4 above.

Stylistic concerns:

There are still multiple spelling/grammar errors.

We have proof read the manuscript again and ran it through a spell checker.

The second-to-last full sentence in the first column of page 2 is confusing and should be reworded. It is the sentence that begins, "At higher amplitude stick-slip dynamics..."

We rearranged the text here and changed this sentence. We hope it is clear now.

Dear reviewer#2,

We thank you for the second, careful reading of our manuscript and we are pleased that you found it improved.

Sincerely,

Per-Anders Thorén

Reviewers' comments:

Reviewer #1 (Remarks to the Author):

The authors have comprehensively address the concerns raised in the previous review. As they state, there are some aspects to the study that merit further investigation but it is reasonable to leave these to a future study. The referee encourages the authors to pursue these points in the future (e.g., the response at a step, the effect of temperature, and the verification of with single lattice slip events).